IPPP/23/18, DCPT/23/36, DESY-23-038

# `HEJ` 2.2: W boson pairs and Higgs boson plus jet production at high energies

Jeppe R. Andersen[a], Bertrand Ducloué[b], Conor Elrick[b], Hitham Hassan[a], Andreas Maier[c], Graeme Nail[b], Jérémy Paltrinieri[b], Andreas Papaefstathiou[d], Jennifer M. Smillie[b]

[a]*Institute for Particle Physics Phenomenology, University of Durham, Durham, DH1 3LE, UK*
[b]*Higgs Centre for Theoretical Physics, University of Edinburgh, Peter Guthrie Tait Road, Edinburgh, EH9 3FD, UK*
[c]*Deutsches Elektronen-Synchrotron DESY, Platanenallee 6, 15738 Zeuthen, Germany*
[d]*Department of Physics, Kennesaw State University, Kennesaw, GA 30144, USA*

## Abstract

We present version 2.2 of the *High Energy Jets* (`HEJ`) Monte Carlo event generator for hadronic scattering processes at high energies. The new version adds support for two further processes of central phenomenological interest, namely the production of a W boson pair with equal charge together with two or more jets and the production of a Higgs boson with at least one jet. Furthermore, a new prediction for charged lepton pair production with high jet multiplicities is provided in the high-energy limit. The accuracy of `HEJ` 2.2 can be increased further through an enhanced interface to standard predictions based on conventional perturbation theory. We describe all improvements and provide extensive usage examples. `HEJ` 2.2 can be obtained from https://hej.hepforge.org.

*Keywords:* Collider Physics; Monte Carlo Event Generation; Resummation

## NEW VERSION PROGRAM SUMMARY

*Program Title:* `HEJ`.
*Licensing provisions:* GPLv2 or later.
*Programming language:* C++.
*Journal reference of previous version:* Comput.Phys.Commun. 278 (2022) 108404.
*Does the new version supersede the previous version?:* Yes.
*Reasons for the new version:* Support for further scattering processes and improved combination with fixed-order predictions.
*Summary of revisions:* The new release adds the ability to predict the QCD component in the high-energy production of two leptonically decaying W bosons with equal charge, together with two or more jets. High-energy resummation is now also implemented for the production of a Higgs boson together with a single jet, whereas before only processes involving at least two jets had been considered. Pure resummed predictions for lepton pair production via a virtual photon or Z boson together with jets are now provided for high jet multiplicities, where fixed-order matching is no longer feasible. The interface to fixed-order generators has been extended significantly, including options for differential reweighting to next-to-leading order, filtering of jets with low transverse momentum, and the capability to stream a wider range of input event

files.

*Nature of problem:* Hadronic scattering processes at high energies, i.e. with large invariant masses between jets, are of great phenomenological interest. This is in particular the case for measurements of weak-boson scattering and weak-boson fusion production of a Higgs boson. In the high-energy region, standard perturbation theory exhibits poor convergence for the QCD contributions, which limits the predictive power of conventional Monte Carlo generators.

*Solution method:* The poor convergence of the perturbative series can be traced to the appearance of large high-energy logarithms. `HEJ` is a fully flexible event generator combining fixed-order accuracy with the all-order resummation of such logarithms, based on the *High Energy Jets* framework. The new version `HEJ` 2.2 provides accurate predictions for a range of processes of central phenomenological interest, including the QCD component of same-sign W boson pair production with multiple jets and Higgs boson production in association with one or more jets.

## Contents

## 1. Introduction

Hadronic scattering processes in the high-energy region are of great phenomenological interest. Prime examples include coupling measurements in weak boson fusion and weak boson scattering. To suppress the background in these measurements, one typically requires jets with large invariant masses and a large difference in rapidity. These requirements strongly enhance the contribution from the high-energy region, which is characterised by invariant masses that are much larger than all transverse scales, or equivalently large rapidity separations with no strong hierarchy between the transverse momenta. In this kinematic regime, high-energy logarithms of the large ratio $\hat{s}/p_\perp^2$ arise in perturbation theory, where $\hat{s}$ is the square of the partonic centre-of-mass energy and $p_\perp$ the characteristic transverse momentum scale. In the QCD component, these large logarithms jeopardise the convergence of the perturbative series.

*High Energy Jets* (`HEJ`) is both a framework and a flexible Monte Carlo generator for the all-order resummation of high-energy logarithms [1, 2, 3]. In `HEJ` 2 [4], this high-energy resummation can additionally be matched to leading-order predictions obtained using conventional fixed-order generators [5]. `HEJ` has been validated against data in experimental studies of pure multijet production [6, 7, 8, 9], lepton pair production via a virtual W boson, photon, or Z boson in association with two or more jets [10, 11, 12, 13], and Higgs boson production with jets [14].

In the following, we present `HEJ` 2.2. This new version implements high-energy resummation for the production of two leptonically decaying W bosons with the same charge in association with two or more jets. Moreover, the existing implementation for the production of a Higgs boson with jets has been extended to also cover Higgs boson production with a single jet. Resummation for the production of a charged lepton pair via a virtual photon or Z boson together with two or more jets is now also supported for higher multiplicities where no fixed-order prediction is available. Furthermore, new options have been added, for instance to facilitate differential next-to-leading-order matching and to separate events with soft jets. In section 2, we briefly summarise *High Energy Jets* and describe the various improvements in version 2.2. We give examples for the use of the new features in section 3 and conclude in section 4. Appendix A contains instructions for download and installation of `HEJ` 2.2.

## 2. Features of `HEJ` 2.2

### 2.1. `HEJ` *in a nutshell*

Before describing the changes in `HEJ` 2.2, let us briefly review the general formalism and program structure. As input, `HEJ` requires leading-order (LO) events, generated with e.g. SHERPA [15] or MADGRAPH5_AMC@NLO [16]. For higher jet multiplicities exact fixed-order generation becomes increasingly time consuming. To address this problem, `HEJ` includes the fast `HEJ` fixed-order generator `HEJFOG` based on the high-energy approximation of the leading-order matrix elements.

Using the kinematics of each (approximate or exact) input event, we identify whether resummation is possible. For each event that permits resummation, `HEJ` generates a number of matching events in the resummation phase space, which include real and virtual corrections to all orders in the high-energy limit. Details are given in [17]. Together with the unchanged non-resummable input events, the generated resummation events are then passed on to any number of output event files and/or analyses. This standard control flow is depicted in figure 1. It can be modified through `HEJ` options, such that e.g. non-resummable events are discarded.

The first type of event kinematics for which resummation is implemented are leading-logarithmic (LL) configurations, which for pure multijet production have to fulfil the following constraints:

1. The flavour of the most backward outgoing parton has to match the flavour of the backward incoming parton.
2. The flavour of the most forward outgoing parton has to match the flavour of the forward incoming parton.

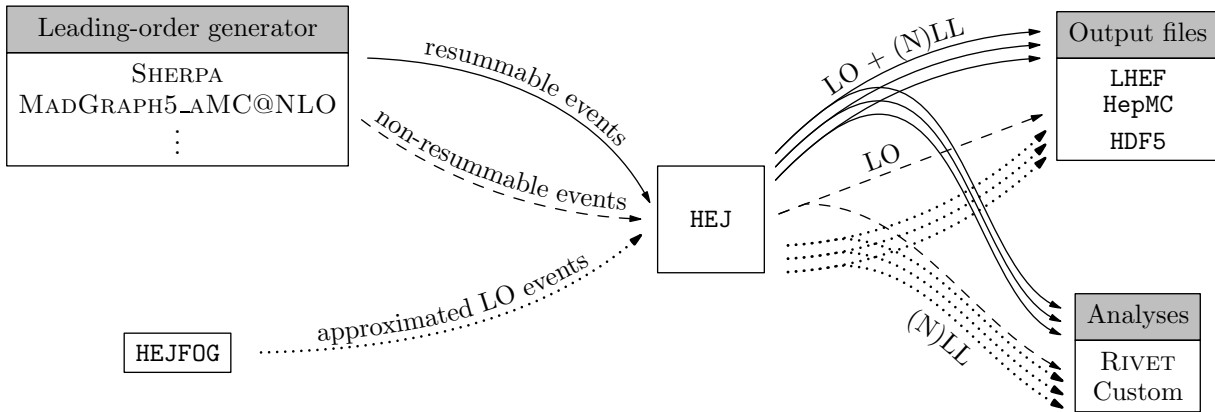

Figure 1: Standard `HEJ` control flow.

3. All other outgoing partons have to be gluons.

These criteria remain the same in processes involving virtual photons and/or Z bosons. For virtual W bosons, the incoming and outgoing flavours in criteria 1 and 2 only have to match up to the change induced by W boson couplings. In the case of a final-state Higgs boson, configurations where the backward (forward) incoming parton is a quark or antiquark and the most backward (forward) outgoing particle is the Higgs boson are formally subleading. Nevertheless, we also implement resummation for such configurations. Depending on the process, resummation is also implemented for two further types of next-to-leading-logarithmic (NLL) configurations. These configurations differ from LL ones as follows.

- *Unordered gluon:* Either the most forward or most backward outgoing parton is a gluon, and the next outgoing parton in rapidity order is a quark or antiquark whose flavour matches the one of the respective incoming parton.

- *Quark-antiquark:* A pair of final-state gluons that are adjacent in rapidity is replaced by a quark-antiquark pair.

The current status of the implemented resummation is summarised in table 1.

The resummation events generated for the LL and supported NLL configurations are given a final matrix element weight of

$$|\mathcal{M}_{\text{HEJ}}|^2 \frac{|\mathcal{M}_{\text{LO}}|^2}{|\mathcal{M}_{\text{HEJ,LO}}|^2}, \tag{1}$$

where $\mathcal{M}_{\text{HEJ}}$ is the all-order scattering matrix element in the high-energy approximation, $\mathcal{M}_{\text{HEJ,LO}}$ its leading-order truncation, calculated for the kinematics of the input event and $|\mathcal{M}_{\text{LO}}|^2$ is taken from the LO input.

To illustrate the structure of the `HEJ` matrix element, we first focus on LL configurations in pure multijet production. We denote these configurations as $f_a f_b \to f_a \cdots f_b$, where $f_a$ is the flavour of the incoming parton in the backward direction with momentum $p_a$. Correspondingly, we use $f_b$ and $p_b$ for the flavour and momentum of the forward incoming parton. The final state contains $n$ partons with momenta $p_1, \ldots, p_n$, which we

| Process | pure LL | LO + LL | NLL | |
| --- | --- | --- | --- | --- |
| | | | unordered gluon | quark-antiquark |
| $\geq 2$ jets | HEJ 2.0 | HEJ 2.0 | HEJ 2.0 | HEJ 2.1 |
| H + 1 jet | — | HEJ 2.2 | N/A | N/A |
| H + $\geq 2$ jets | HEJ 2.0 | HEJ 2.0 | HEJ 2.0 | — |
| W + $\geq 2$ jets | HEJ 2.1 | HEJ 2.1 | HEJ 2.1 | HEJ 2.1 |
| $Z/\gamma + \geq 2$ jets | HEJ 2.2 | HEJ 2.1 | HEJ 2.1 | — |
| $W^{\pm}W^{\pm} + \geq 2$ jets | — | HEJ 2.2 | — | — |

Table 1: Implemented processes and higher-order logarithmic corrections in `HEJ`. The "pure LL" column lists processes implemented in the `HEJFOG`. The NLL columns include both pure NLL and NLL matched to LO.

order by rapidity, viz. $y_1 < \cdots < y_n$. The most backward outgoing parton has flavour $f_a$, the most forward one flavour $f_b$, and all other outgoing partons are gluons. This implies that the outgoing parton with flavour $f_a$ has momentum $p_1$ and the outgoing parton with flavour $f_b$ has momentum $p_n$. However, note that this identification does not necessarily hold at NLL. For example, if there is an unordered gluon in the backward direction then the outgoing parton with flavour $f_a$ has momentum $p_2$. Using the introduced notation, we can write the general form of the squared `HEJ` matrix element as

$$
\overline{\left|\mathcal{M}_{\texttt{HEJ}}^{f_a f_b \to f_a \cdots f_b}\right|}^2 = \mathcal{B}_{f_a, f_b}(p_a, p_b, p_1, p_n)
$$
$$
\cdot \prod_{i=1}^{n-2} \mathcal{V}(p_a, p_b, p_1, p_n, q_i, q_{i+1}) \tag{2}
$$
$$
\cdot \prod_{i=1}^{n-1} \mathcal{W}(q_{i\perp}, y_i, y_{i+1}),
$$

where $q_i = p_a - \sum_{j=1}^{i} p_j$ is the $t$-channel momentum after the emission of parton $i$. $\mathcal{B}_{f_a, f_b}$ is derived from the modulus square of the Born-level matrix element for the process $f_a f_b \to f_a f_b$, $\mathcal{V}$ accounts for the real emission of the $n-2$ gluons in between $f_a$ and $f_b$, and $\mathcal{W}$ incorporates the virtual and unresolved real corrections.

The Born-level function $\mathcal{B}_{f_a, f_b}$ is given by

$$
\mathcal{B}_{f_a, f_b}(p_a, p_b, p_1, p_n) = \frac{(4\pi\alpha_s)^n}{4(N_C^2 - 1)} \frac{K_{f_a}}{q_1^2} \frac{K_{f_b}}{q_{n-1}^2} \|S_{f_a f_b \to f_a \cdots f_b}\|^2, \tag{3}
$$

where $\alpha_s$ is the strong coupling constant and $N_C = 3$ the number of colours. $K_{f_a}$ and $K_{f_b}$ are generalised colour factors depending on the respective parton flavour and, in the case of gluons, also the parton momentum. For quarks and antiquarks one finds $K_f = C_F = \frac{N_C^2 - 1}{2N_C}$; the factor $K_g$ for gluons is derived in [2]. $S_{f_a f_b \to f_a \cdots f_b}$ denotes the contraction of two currents:

$$
\|S_{f_a f_b \to f_a \cdots f_b}\|^2 \equiv \|j^a \cdot j^b\|^2 = \sum_{\substack{\lambda_a = +, - \\ \lambda_b = +, -}} |j^{\mu, \lambda_a}(p_1, p_a) j_\mu^{\lambda_b}(p_n, p_b)|^2, \tag{4}
$$

where $j_\mu^\lambda$ is the current

$$j_\mu^\lambda(p,q) = \bar{u}^\lambda(p)\gamma_\mu u^\lambda(q) \tag{5}$$

for helicity $\lambda$. `HEJ` employs the symbolic manipulation language `FORM` [18] to generate compact symbolic expressions for current contractions.

The real corrections are given by contractions of Lipatov vertices [5]:

$$\mathcal{V}(p_a, p_b, p_1, p_n, q_i, q_{i+1}) = -\frac{C_A}{q_i^2 q_{i+1}^2} V_\mu(p_a, p_b, p_1, p_n, q_i, q_{i+1}) V^\mu(p_a, p_b, p_1, p_n, q_i, q_{i+1}),$$

$$\tag{6}$$

$$\begin{aligned}
V^\mu(p_a, p_b, p_1, p_n, q_i, q_{i+1}) = & -(q_i + q_{i+1})^\mu \\
& + \frac{p_a^\mu}{2}\left(\frac{q_i^2}{p_{i+1}\cdot p_a} + \frac{p_{i+1}\cdot p_b}{p_a\cdot p_b} + \frac{p_{i+1}\cdot p_n}{p_a\cdot p_n}\right) + p_a \leftrightarrow p_1 \\
& - \frac{p_b^\mu}{2}\left(\frac{q_{i+1}^2}{p_{i+1}\cdot p_b} + \frac{p_{i+1}\cdot p_a}{p_b\cdot p_a} + \frac{p_{i+1}\cdot p_1}{p_b\cdot p_1}\right) - p_b \leftrightarrow p_n,
\end{aligned} \tag{7}$$

with $C_A = N_C$.

Finally, the virtual and unresolved real corrections $\mathcal{W}$ can be expressed in terms of the regularised Regge trajectory $\omega^0$:

$$\mathcal{W}(q_{j\perp}, y_j, y_{j+1}) = \exp[\omega^0(q_{j\perp})(y_{j+1} - y_j)]. \tag{8}$$

For a detailed discussion and an explicit expression for $\omega^0$ see [17].

The generalisations to NLL configurations and additional non-partonic final state particles are derived in [10, 13, 5, 19, 20, 21, 14]. In all cases one finds a factorisation into a Born-level function, resolved real emissions, and virtual and unresolved real corrections. In the absence of interference, one recovers the same structure as in equation (2). In particular, the functions $\mathcal{V}$ and $\mathcal{W}$ comprising the all-order corrections are universal, whereas the Born-level function $\mathcal{B}$ is process dependent.

### 2.2. High-energy resummation for W pair production

Based on the pure-QCD LL configurations $f_a f_b \to f_a \cdots f_b$, additional W bosons can be produced via emission off the partons $f_a$ and $f_b$. In HEJ 2.2, we consider LL configurations with two leptonically decaying W bosons with equal charge. For definiteness, we discuss configurations $f_a f_b \to (W^- \to e\bar{\nu}_e)(W^- \to \mu\bar{\nu}_\mu)f_{a'} \cdots f_{b'}$, where the rapidities of the final-state charged and neutral leptons do not necessarily respect any rapidity ordering. Note that the couplings to the W bosons induce flavour changes $f_a \to f_{a'}$ and $f_b \to f_{b'}$. We use a diagonal CKM matrix and do not include third-generation quarks/anti-quarks, i.e. the number of active flavours is 4. The production of two positively charged W bosons and the decay of the two W bosons into the same lepton flavours is completely analogous.

We identify two contributions to the amplitude. Parton $f_a$ can either couple to the W boson that decays into an electron and its antineutrino or to the W boson decaying into

a muon and its antineutrino. In the first case, the $t$-channel momenta are given by

$$q_{i,e} = p_a - p_e - p_{\bar{\nu}_e} - \sum_{j=1}^{i} p_j, \tag{9}$$

where $p_e$ is the momentum of the electron and $p_{\bar{\nu}_e}$ the momentum of its antineutrino. In the second case the $t$-channel momenta are

$$q_{i,\mu} = p_a - p_\mu - p_{\bar{\nu}_\mu} - \sum_{j=1}^{i} p_j \tag{10}$$

with the muon momentum $p_\mu$ and the corresponding antineutrino momentum $p_{\bar{\nu}_\mu}$. The resulting modulus square of the matrix element including interference is [21]

$$
\begin{aligned}
\left| \mathcal{M}_{\text{HEJ}}^{f_a f_b \to e \bar{\nu}_e \mu \bar{\nu}_\mu f_{a'} \cdots f_{b'}} \right|^2 &= \frac{(4\pi\alpha_s)^n}{4(N_c^2 - 1)} \, K_{f_a} K_{f_b} C_A^{n-2} \\
&\times \Bigg( \frac{\|j_{W,e}^a \cdot j_{W,\mu}^b\|^2}{q_{1,e}^2 q_{n-1,e}^2} \prod_{i=1}^{n-2} \frac{-V^2(q_{i,e}, q_{i+1,e})}{q_{i,e}^2 q_{i+1,e}^2} \prod_{i=1}^{n-1} \mathcal{W}(q_{i,e\perp}, y_i, y_{i+1}) \\
&+ \frac{\|j_{W,\mu}^a \cdot j_{W,e}^b\|^2}{q_{1,\mu}^2 q_{n-1,\mu}^2} \prod_{i=1}^{n-2} \frac{-V^2(q_{i,\mu}, q_{i+1,\mu})}{q_{i,\mu}^2 q_{i+1,\mu}^2} \prod_{i=1}^{n-1} \mathcal{W}(q_{i,\mu\perp}, y_i, y_{i+1}) \\
&- \frac{2\Re\{(j_{W,e}^a \cdot j_{W,\mu}^b)(\overline{j_{W,\mu}^a \cdot j_{W,e}^b})\}}{\sqrt{q_{1,e}^2 q_{1,\mu}^2} \sqrt{q_{n-1,e}^2 q_{n-1,\mu}^2}} \\
&\times \prod_{i=1}^{n-2} \frac{V(q_{i,e}, q_{i+1,e}) \cdot V(q_{i,\mu}, q_{i+1,\mu})}{\sqrt{q_{i,e}^2 q_{i,\mu}^2} \sqrt{q_{i+1,e}^2 q_{i+1,\mu}^2}} \prod_{i=1}^{n-1} \mathcal{W}(\sqrt{q_{i,e\perp} q_{i,\mu\perp}}, y_i, y_{i+1}) \Bigg).
\end{aligned}
\tag{11}
$$

Here, we have introduced contractions between generalised currents $j_{W,l}^c$ accounting for the coupling between a parton with flavour $f_c$ and a W boson decaying into a charged lepton $l$ and the corresponding antineutrino. The contractions are

$$\|j_{W,e}^a \cdot j_{W,\mu}^b\|^2 = \left| j_V^{\rho, \lambda_a \lambda_e}(p_1, p_a, p_e, p_{\bar{\nu}_e}) j_V^{\sigma, \lambda_b \lambda_\mu}(p_{n+2}, p_b, p_\mu, p_{\bar{\nu}_\mu}) g_{\rho\sigma} \right|^2, \tag{12}$$

$$\|j_{W,\mu}^a \cdot j_{W,e}^b\|^2 = \left| j_V^{\rho, \lambda_a \lambda_\mu}(p_1, p_a, p_\mu, p_{\bar{\nu}_\mu}) j_V^{\sigma, \lambda_b \lambda_e}(p_{n+2}, p_b, p_e, p_{\bar{\nu}_e}) g_{\rho\sigma} \right|^2, \tag{13}$$

$$
\begin{aligned}
(j_{W,e}^a \cdot j_{W,\mu}^b)(\overline{j_{W,\mu}^a \cdot j_{W,e}^b}) &= j_V^{\rho, \lambda_a \lambda_e}(p_1, p_a, p_e, p_{\bar{\nu}_e}) j_V^{\sigma, \lambda_b \lambda_\mu}(p_{n+2}, p_b, p_\mu, p_{\bar{\nu}_\mu}) g_{\rho\sigma} \\
&\times \overline{j_V^{\alpha \lambda_a \lambda_\mu}(p_1, p_a, p_\mu, p_{\bar{\nu}_\mu}) j_V^{\beta, \lambda_b \lambda_e}(p_{n+2}, p_b, p_e, p_{\bar{\nu}_e}) g_{\alpha\beta}},
\end{aligned}
\tag{14}
$$

where the parton helicities $\lambda_a$ and $\lambda_b$ are determined by the flavour of the respective parton, namely $\lambda_c = -$ if $f_c$ is a quark and $\lambda_c = +$ if $f_c$ is an antiquark. The electron helicity $\lambda_e$ and the muon helicity $\lambda_\mu$ correspond to the charge sign of the parent W boson, i.e. $\lambda_e = \lambda_\mu = -$ in the present case. We have introduced a generalised current $j_V^{\rho, \lambda_a \lambda_\ell}$ for

the coupling of a parton with helicity $\lambda$ to a leptonically decaying vector boson[1] $V$ with lepton helicity $\lambda_\ell$. It is given by [20]

$$
\begin{aligned}
j_V^{\rho,\lambda_a\lambda_\ell}(p_1,p_a,p_\ell,p_{\bar\nu_\ell}) &= \frac{g_V^2}{2}\frac{1}{p_V^2-M_V^2+i\,\Gamma_V M_V}\,\bar{u}^{\lambda_\ell}(p_\ell)\gamma_\alpha v^{\lambda_\ell}(p_{\bar\nu_\ell}) \\
&\cdot\left(\frac{\bar{u}^{\lambda_a}(p_1)\gamma^\alpha(\slashed{p}_V+\slashed{p}_1)\gamma^\rho u^{\lambda_a}(p_a)}{(p_V+p_1)^2}+\frac{\bar{u}^{\lambda_a}(p_1)\gamma^\rho(\slashed{p}_a-\slashed{p}_V)\gamma^\alpha u^{\lambda_a}(p_a)}{(p_a-p_V)^2}\right).
\end{aligned}
\tag{15}
$$

$p_V=p_\ell+p_{\bar\nu_\ell}$ is the vector boson momentum, $g_V$ its coupling to the fermion $f_a$, $M_V$ its mass, and $\Gamma_V$ the width.

### 2.3. Higgs boson production with a single jet

In the gluon-fusion production of a Higgs boson together with one or more jets new LL configurations beyond those derived from pure multijet production (c.f. section 2.1) arise. In these configurations, one of the incoming partons is a gluon while the corresponding most forward or most backward outgoing particle is the Higgs boson, i.e. $gf_b \to H\cdots f_b$ or $f_a g \to f_a \cdots H$. Without loss of generality we consider the former configuration. The modulus square of the HEJ matrix element reads [14]

$$
\begin{aligned}
\overline{\left|\mathcal{M}_{\mathrm{HEJ}}^{gf_b\to H\cdots f_b}\right|}^2 &= \mathcal{B}_{H,f_b}(p_a,p_b,p_1,p_n) \\
&\cdot\prod_{i=1}^{n-2}\mathcal{V}(p_a,p_b,p_a,p_n,q_i,q_{i+1}) \\
&\cdot\prod_{i=1}^{n-1}\mathcal{W}(q_{i\perp},y_i,y_{i+1}),
\end{aligned}
\tag{16}
$$

with the universal real and virtual correction factors $\mathcal{V}$ and $\mathcal{W}$ defined in equations (6) and (8). The only differences to the pure QCD case in equation (2) are the replacement $p_1 \to p_a$ in the third argument of $\mathcal{V}$ and the adjustment of the process-dependent Born function to [14]

$$
\mathcal{B}_{H,f_b} = \frac{(4\pi\alpha_s)^{n-1}}{4(N_c^2-1)}\frac{1}{q_1^2}\frac{K_{f_b}}{q_{n-1}^2}\left\|S_{gf_b\to Hf_b}\right\|^2,
\tag{17}
$$

$$
\left\|S_{gf_b\to Hf_b}\right\|^2 = \sum_{\substack{\lambda_a=+,-\\\lambda_b=+,-}}\left|\epsilon_\mu^{\lambda_a}(p_a)\,V_H^{\mu\nu}(p_a,p_a-p_1)\,j_\nu^{\lambda_b}(p_n,p_b)\right|^2,
\tag{18}
$$

where $\epsilon^{\lambda_a}(p_a)$ is the polarisation vector of the incoming gluon and $V_H$ the effective vertex coupling the Higgs boson to two gluons at one-loop, including finite quark-mass dependence. The structure of equation (16) then allows finite quark-mass dependence to be applied for arbitrarily high numbers of jets.

---

[1]Here, we assume a W boson. However, the same expression holds for neutral vector bosons after replacing the antineutrino momentum $p_{\bar\nu_\ell}$ by the antilepton momentum $p_{\bar\ell}$.

## 2.4. Spillover from small transverse momenta

As described in section 2.1, a number of all-order resummation events are generated for each resummable input event. Since the resummation events include real corrections, the resulting kinematics differ slightly from the kinematics of the input events. While jet rapidities are always preserved, this is generally neither true for transverse momenta nor for the rapidities of any other particles. This implies that cuts imposed on the leading-order generation should be significantly looser than the final analysis cuts. Empirically, the difference in transverse momentum cuts should be about 10-20%, with a slight increase towards larger multiplicities.

Simply adjusting the cuts in the leading-order generation is correct, but inefficient: events with small transverse momenta dominate the leading-order prediction, but only give a small contribution to the final resummed results. It is therefore more efficient to split up the leading-order generation. One first generates a high-statistics sample in which all particles fulfil the transverse momentum cuts of the analysis. Then, one generates a second low-statistics sample where in each event there is at least one particle with small transverse momentum that does not pass the final cuts. Since the two samples are disjoint, one can separately apply `HEJ` resummation to each sample and add up the results.

However, implementing the requirement of at least one "soft" particle is often not straightforward with standard fixed-order generators. To facilitate resummation for the low transverse momentum sample, `HEJ` 2.2 introduces a new option for discarding events in which all jets are above the analysis threshold. An example is given in section 3.1.3.

## 2.5. Matching to Next-to-Leading Order

To improve the accuracy of the obtained total cross section to next-to-leading order (NLO), one can obviously multiply the `HEJ` prediction by a flat factor of $\sigma_{\mathrm{NLO}}/\sigma_{\mathtt{HEJ}}$, where $\sigma_{\mathtt{HEJ}}$ is the leading-order accurate total cross section according to `HEJ` and $\sigma_{\mathrm{NLO}}$ the total cross section at NLO. `HEJ` 2.2 enables us to achieve NLO accuracy also in differential distributions. Considering a distribution $d\sigma/d\mathcal{O}$ in some observable $\mathcal{O}$, we can combine NLO and `HEJ` resummation through the reweighting

$$\left(\frac{d\sigma}{d\mathcal{O}}\right)_{\mathtt{HEJ}+\mathrm{NLO}} = \frac{(d\sigma/d\mathcal{O})_{\mathrm{NLO}}}{(d\sigma/d\mathcal{O})_{\mathtt{HEJ},\mathrm{NLO}}} \left(\frac{d\sigma}{d\mathcal{O}}\right)_{\mathtt{HEJ}}. \tag{19}$$

Here, the subscript `HEJ` denotes the prediction before reweighting, NLO the NLO-accurate prediction, and `HEJ`,NLO the truncation of the `HEJ` prediction to NLO. In section 3.2.2, we show in an example how to truncate the `HEJ` prediction and obtain NLO-reweighted distributions.

## 2.6. Predictions without Fixed-Order Matching

The computational cost for generating the fixed-order input events rises steeply with the jet multiplicity. For this reason, `HEJ` includes the `HEJFOG`, a fast generator based on the leading-order truncation of the `HEJ` matrix element given in equation (2). The intended use is that one will generate exact low-multiplicity input events with a conventional generator and supplement them with approximate high-multiplicity events using the `HEJFOG`. In `HEJ` 2.2, the `HEJFOG` includes charged lepton pair production with jets as a new process.

Furthermore, the generation efficiency for the production of a W boson with jets has been improved by aligning the rapidity of the W boson with its emitter, reducing the Monte Carlo uncertainty by a factor of up to 2.

## 3. Example Usage

In the following, we show how the new features in `HEJ` 2.2 can be used in practice. For concreteness, we will generate leading-order events with SHERPA 2.2 and analyse the output with RIVET 3 [22]. However, we stress that any leading-order generator that can produce event files in the LHEF format [23] is supported. In addition to the direct RIVET interface, `HEJ` can write its output to event files in various formats and allows arbitrary custom analyses via plugins. Since these options are not new, we refer to the `HEJ` documentation on `https://hej.hepforge.org` for details.

### 3.1. Same-sign W pair production with jets

We first consider the process $pp \to (W^- \to e\bar{\nu}_e)(W^- \to \mu\bar{\nu}_\mu) + \geq 2$ jets with the parameters shown in table 2. $H_T = p_{e\perp} + p_{\bar{\nu}_e\perp} + p_{\mu\perp} + p_{\bar{\nu}_\mu\perp} + \sum_{i=1}^{n} p_{i\perp}$ is the sum of the scalar transverse momenta of all outgoing particles.

| | |
|---|---|
| Collider energy | $\sqrt{s} = 13\,\text{TeV}$ |
| Scales | $\mu_r = \mu_f = \frac{H_T}{2}$ |
| PDF set | CT14nlo |
| Electroweak input parameters | $\alpha = 1/132.3572$ |
| | $m_W = 80.385\,\text{GeV}$ |
| | $\Gamma_W = 2.085\,\text{GeV}$ |
| | $m_Z = 91.1876\,\text{GeV}$ |
| | $\Gamma_Z = 2.4952\,\text{GeV}$ |
| Jet definition | anti-$k_t$ [24] |
| | $R = 0.4$ |
| | $p_\perp > 20\,\text{GeV}$ |

Table 2: Parameters for the production of multiple jets together with a same-sign W boson pair decaying to charged leptons and neutrinos.

### 3.1.1. Generating leading-order input

To produce the required leading-order input, we can use SHERPA with the following runcard.

Run.dat

```
(run){
  EVENTS 10000

  EVENT_OUTPUT LHEF[events_WW2j]
```

```
  # theory parameters
  EW_SCHEME 3
  1/ALPHAQED(0) 132.3572
  USE_PDF_ALPHAS 1

  # Z boson mass and width
  MASS[23] 91.1876
  WIDTH[23] 2.4952

  # W boson mass and width
  MASS[24] 80.385
  WIDTH[24] 2.085

  # massless charm quark
  MASSIVE[4] 0
  YUKAWA[4] 0.
  MASS[4] 0.

  # massless bottom quark
  MASSIVE[5] 0
  YUKAWA[5] 0.
  MASS[5] 0.

  # collider beam
  BEAM_ENERGY:=6500.
  BEAM_1 2212 BEAM_ENERGY
  BEAM_2 2212 BEAM_ENERGY

  # PDF
  PDF_LIBRARY LHAPDFSherpa
  PDF_SET CT14nlo

  # Set square of renormalisation and factorisation scale
  SCALES VAR{H_T2/4}

  EVENT_GENERATION_MODE Weighted
  ME_SIGNAL_GENERATOR Comix

  # disable everything beyond fixed order
  FRAGMENTATION Off
  YFS_MODE 0
  MI_HANDLER None
  SHOWER_GENERATOR None
  CSS_MAXEM 0
  BEAM_REMNANTS 0
}(run)

(processes){
  Process 93 93 -> 93 93 11 -12 13 -14
  Order (*,4)
  End process
}(processes)

(selector){
  # require 2 anti-kt jets with pt > 15 GeV and R = 0.4
  FastjetFinder antikt 2 15. 0.0 0.4
}(selector)
```

The various settings are explained in more detail in the SHERPA manual. Since HEJ

treats all quarks as massless, we have to set the charm and bottom quark masses to zero for consistency. As explained in section 2.4, leading-order events containing particles with transverse momenta below the analysis cuts can still contribute to the resummed prediction. For this reason, we accept jets with transverse momenta as low as 15 GeV, despite having an analysis cut of 20 GeV. In section 3.1.3, we will discuss a computationally more efficient way to incorporate this contribution from leading-order events that do not pass the analysis transverse momentum cuts.

We can now generate input events for the case of two jets by running

```
Sherpa
```

in the directory containing `Run.dat`. With the SHERPA 2.2.15 installation inside the `HEJ` Docker image this yields a cross section of $(1.88795 \pm 0.109611)$ fb. Results may differ between versions and when re-using integration grids from previous SHERPA runs. We should then also produce event files for higher jet multiplicities after adjusting the `EVENT_OUTPUT`, `Process`, and `FastjetFinder` entries in `Run.dat`. Increasing the number of jets to three leads to a cross section of $(1.47689 \pm 0.10331)$ fb.

To avoid the creation of large intermediate event files, we can use named pipes instead, i.e. we run SHERPA with

```
mkfifo events_WW2j.lhe
Sherpa &
```

Previous `HEJ` versions only accepted input from pipes if the total cross section was equal to the sum of event weights, which is not the case for SHERPA event files. This restriction is lifted in the new version 2.2, which removes a potential bottleneck in time and storage.

*3.1.2.* `HEJ` *resummation*

In addition to the leading-order event input, `HEJ` needs a configuration file. Adapting the template `config.yml` distributed with the `HEJ` source code to the parameters listed in table 2 we get

config_WWjets.yml

```
## Number of attempted resummation phase space points for each input event
trials: 10

resummation jets:        # resummation jet properties
  min pt: 20             # minimum jet transverse momentum
  algorithm: antikt      # jet clustering algorithm
  R: 0.4                 # jet R parameter

fixed order jets:        # properties of input jets
  min pt: 15
  # by default, algorithm and R are like for resummation jets

## Treatment of the various event classes
## the supported settings are: reweight, keep, discard
## non-resummable events cannot be reweighted
event treatment:
  FKL: reweight # enable LL resummation
  # NLL resummation is not implemented for this process,
  # so we keep all other events as they are
```

```
  unordered: keep
  extremal qqbar: keep
  central qqbar: keep
  non-resummable: keep

## Central scale choice or choices
scales: H_T/2

## Analyses
analyses:
  - rivet: [MC_WWINC, MC_WWJETS]  # rivet analysis names
    output: WW2j # name of the yoda files, ".yoda" and scale suffix will be added

## Selection of random number generator and seed
random generator:
  name: mixmax
  seed: 1

## Vacuum expectation value
vev: 246.2196508

## Properties of the weak gauge bosons
particle properties:
  Higgs:
    mass: 125
    width: 0.004165
  W:
    mass: 80.385
    width: 2.085
  Z:
    mass: 91.1876
    width: 2.4952

## Whether or not to include higher order logs
log correction: false
```

Here, we choose to pass the resummed events directly to the RIVET analyses[2] MC_WWINC and MC_WWJETS. Using the Docker virtualisation software, we can run HEJ [25] with the following command.

```
docker run -v $PWD:$PWD -w $PWD hejdock/hej HEJ config_WWjets.yml events_WW2j.lhe
```

Alternatively, after compiling and installing HEJ and its dependencies we can use

```
HEJ config_WWjets.yml events_WW2j.lhe
```

HEJ outputs a cross section of $(1.094 \pm 0.06614)$ fb. It also produces the RIVET analysis output file WW2j.yoda. We produce resummed predictions for the higher-multiplicity event files events_WWnj.lhe in the same way, after changing the analyses entry in config_WWjets.yml to

```
analyses:
  - rivet: [MC_WWINC, MC_WWJETS]
    output: WWnj
```

---

[2]The MC_WWINC and MC_WWJETS analyses are written for opposite-sign W boson pair production but can also be used for same-sign W boson pairs.

and adjusting the event file name when running `HEJ`. To guarantee statistically independent output, it is also recommended to change the `seed` entry for each run. If we nonetheless use the default seed with the previously generated event file `events_WW3j.lhe` we obtain a cross section of $(0.905 \pm 0.07534)$ fb.

We then combine the results for the different jet multiplicities with

```
yodastack -o WWjets.yoda WW*j.yoda
```

and produce plots with

```
rivet-mkhtml WWjets.yoda
```

As examples, we show the inclusive jet multiplicities and the distribution of the rapidity difference between the two W bosons obtained from resumming fixed-order predictions with two and three jets in figure 2. In contrast to the recommended usage, we have not modified the random number generator seeds between runs in order to facilitate reproducing this figure.[3] We observe that the bulk of the events does not pass the analysis cuts.

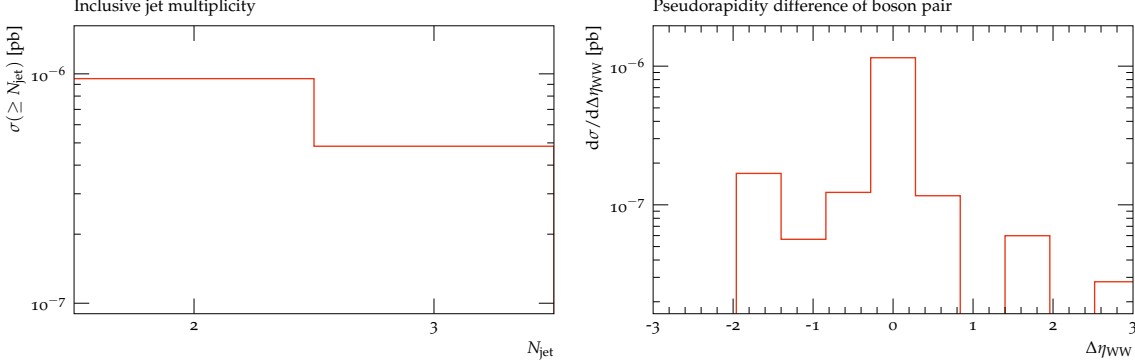

Figure 2: Inclusive $N$-jet cross sections (left) and rapidity difference between the two W bosons (right) obtained with SHERPA and `HEJ` 2.2 for the production of two leptonically decaying W$^-$ bosons with at least two jets.

### 3.1.3. Dedicated low transverse momentum runs

So far, we have generated the leading-order events with significantly looser transverse momentum cuts than wanted for the final analysis. As argued in section 2.4, it is more efficient to split the generation into a high-statistics run with the strict cuts used in the final analysis and a low-statistics run with loose cuts where in each leading-order event there is at least one particle that does not pass the final cuts. For the jet transverse momentum cuts, this separation is facilitated by a new option in `HEJ` 2.2 which ensures the presence of at least one "soft" jet in the input for the low-statistics run. Note that this option only refers to jets; any other particles should be generated according to the loose transverse momentum cuts in both samples.

In detail, one should go through the following steps for the present example of same-sign W pair production with jets:

---

[3]We have slightly changed the plot ranges compared to the default `rivet-mkhtml` settings.

1. Generate the high-statistics sample.

   (a) Change the minimum transverse momentum in the `FastjetFinder` entry in `Run.dat` from 15 to 20.

   (b) Correspondingly, change

   ```
   fixed order jets:
     min pt: 15
   ```

   to

   ```
   fixed order jets:
     min pt: 20
   ```

   in `config_WWjets.yml`

   (c) Run SHERPA and HEJ as before.

   (d) Revert the changes to both configuration files.

2. Generate the low-statistics sample.

   (a) Reduce the number of events generated by Sherpa, for example by setting the `EVENTS` entry in `Run.dat` to 1000.

   (b) Add the entry

   ```
   require low pt jet: true
   ```

   to `config_WWjets.yml`.

   (c) In the `event treatment` entry in `config_WWjets.yml`, change all `keep` values to `discard`. Specifically, change

   ```
   event treatment:
     FKL: reweight
     unordered: keep
     extremal qqbar: keep
     central qqbar: keep
     non-resummable: keep
   ```

   to

   ```
   event treatment:
     FKL: reweight
     unordered: discard
     extremal qqbar: discard
     central qqbar: discard
     non-resummable: discard
   ```

   (d) Change the name of the output file, e.g.

   ```
   analyses:
     - rivet: [MC_WWINC, MC_WWJETS]
       output: WW2j_lowpt
   ```

   (e) Run SHERPA and HEJ.

After repeating these steps for higher jet multiplicities, the samples can again be combined with

```
yodastack -o WWjets.yoda WW*j.yoda WW*j_lowpt.yoda
```

to reproduce the results obtained in section 3.1.2 with better statistics. Example plots resulting from

```
rivet-mkhtml WWjets.yoda
```

with adjusted axis ranges are shown in figure 3.

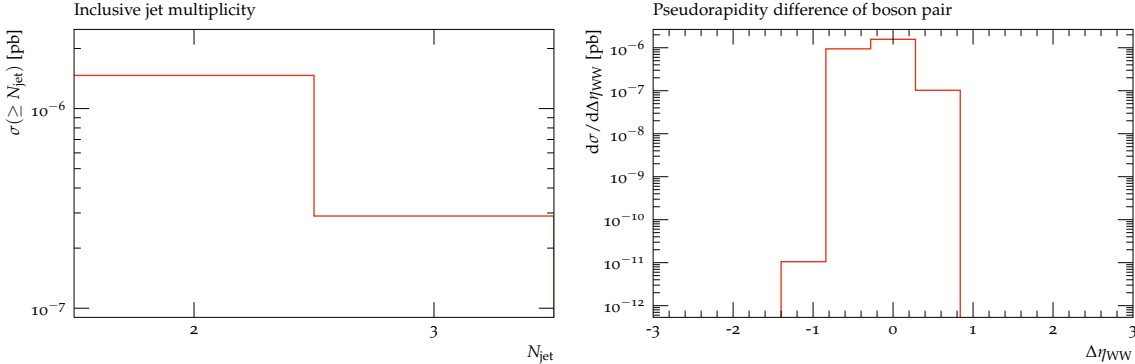

Figure 3: Inclusive $N$-jet cross sections (left) and rapidity difference between the two W bosons (right) combining dedicated high and low transverse momentum runs.

The cross sections for the various runs are summarised in table 3. Here, "low $p_\perp$" refers to the contribution from the phase-space region where at least one jet is softer than 20 GeV. Conversely, "high $p_\perp$" implies that all jets are harder than 20 GeV. Since we do not impose an upper limit on the transverse momenta of the jets in the SHERPA runcard, the low-statistics run covers both of these regions. The "high $p_\perp$" contribution is then removed when running `HEJ` with the `require low pt jet` option, which requires that at least one jet in the fixed-order input is below the analysis cut. While the contributions from the low transverse momentum run are negligible in this example, they can become relevant in high-statistics distributions that probe the phase space near the minimum jet transverse momentum. What is more, the impact tends to increase with the jet multiplicity.

| Run | | $W^-W^- + 2$ jets | $W^-W^- + 3$ jets |
|---|---|---|---|
| SHERPA | high $p_\perp$ | $1.67517 \pm 0.0702545$ | $1.4744 \pm 0.138443$ |
| | low + high $p_\perp$ | $2.3568 \pm 0.509278$ | $1.70446 \pm 0.387036$ |
| HEJ | high $p_\perp$ | $1.048 \pm 0.05616$ | $1.191 \pm 0.1316$ |
| | low $p_\perp$ ($\times 10^3$) | $33.07 \pm 42.70$ | $1.018 \pm 1.024$ |

Table 3: Reference cross sections in femtobarn when using dedicated low transverse momentum runs. All runs use the default random number generator seeds and the code versions in the `HEJ` Docker image. SHERPA integration grids (`Results.db`) should be deleted between runs to reproduce the exact numbers.

### 3.2. Higgs boson production with one or more jets

We now consider the production of a Higgs boson together with one or more jets. We use the parameters listed in table 4. For the sake of simplicity, we first consider the limit of an infinitely heavy top quark.

| Collider energy | $\sqrt{s} = 13\,\text{TeV}$ |
| --- | --- |
| Scales | $\mu_r = \mu_f = \frac{H_T}{2}$ |
| PDF set | CT14nlo |
| Jet definition | anti-$k_t$ |
| | $R = 0.4$ |
| | $p_\perp > 20\,\text{GeV}$ |

Table 4: Parameters for the production of a Higgs boson together with at least one jet.

In close analogy with section 3.1, we first generate leading-order input events for Higgs boson production with a single jet. We use SHERPA with the following run card

Run.dat

```
(run){
  EVENTS 10000

  EVENT_OUTPUT LHEF[events_H1j]

  # theory parameters
  MODEL HEFT
  USE_PDF_ALPHAS 1
  MASS[25] 125
  WIDTH[25] 0.004165

  # massless charm quark
  MASSIVE[4] 0
  YUKAWA[4] 0.
  MASS[4] 0.

  # massless bottom quark
  MASSIVE[5] 0
  YUKAWA[5] 0.
  MASS[5] 0.

  # collider beam
  BEAM_ENERGY:=6500.
  BEAM_1 2212 BEAM_ENERGY
  BEAM_2 2212 BEAM_ENERGY

  # PDF
  PDF_LIBRARY LHAPDFSherpa
  PDF_SET CT14nlo

  # Set square of renormalisation and factorisation scale
  SCALES VAR{H_T2/4}

  EVENT_GENERATION_MODE Weighted
  ME_SIGNAL_GENERATOR Comix

  # disable everything beyond fixed order
  FRAGMENTATION Off
  YFS_MODE 0
  MI_HANDLER None
```

```
  CSS_MAXEM 0
  BEAM_REMNANTS 0
}(run)

(processes){
  Process 93 93 -> 25 93
  Order (*,0,1)
  End process
}(processes)

(selector){
  FastjetFinder antikt 1 15. 0 0.4
}(selector)
```

For the resummation, we use a similar HEJ configuration file as before. Anticipating further runs with higher multiplicities, we enable resummation for the supported NLL configurations. In the present case this concerns configurations involving an unordered gluon, which first contribute to Higgs boson plus dijet production, cf. section 2.1. Since there is no standard RIVET analysis for stable Higgs boson production, we use the generic MC_JETS analysis.

config_Hjets.yml

```
trials: 10

resummation jets:
  min pt: 20
  algorithm: antikt
  R: 0.4

fixed order jets:
  min pt: 15

event treatment:
  FKL: reweight
  unordered: reweight
  extremal qqbar: keep
  central qqbar: keep
  non-resummable: keep

scales: H_T/2

analyses:
  - rivet: MC_JETS
    output: H1j

vev: 246.2196508

particle properties:
  Higgs:
    mass: 125
    width: 0.004165
  W:
    mass: 80.385
    width: 2.085
  Z:
    mass: 91.1876
    width: 2.4952
```

```
random generator:
  name: mixmax
  seed: 1

log correction: false
```

As described in section 3.1, we then add predictions for higher jet multiplicities. We show reference cross sections in table 5 and example distributions in figure 4. We stress that, as in most examples shown here, the SHERPA and HEJ cross sections are not directly comparable. This is because the SHERPA runcards use a lower transverse momentum cut than HEJ, for the reason explained in section 2.4. Therefore a significant fraction of the input events will not pass the cuts after resummation (see section 3.1.3 for an example on how to deal with this in a more efficient way).

| Run | $H + 1$ jet | $H + 2$ jets | $H + 3$ jets |
|---|---|---|---|
| SHERPA | $24.25 \pm 0.494414$ | $14.0615 \pm 0.469809$ | $6.38016 \pm 0.43697$ |
| HEJ | $11.20 \pm 0.3732$ | $4.420 \pm 0.1953$ | $1.657 \pm 0.1558$ |

Table 5: Reference cross sections in picobarn for Higgs boson plus jet production.

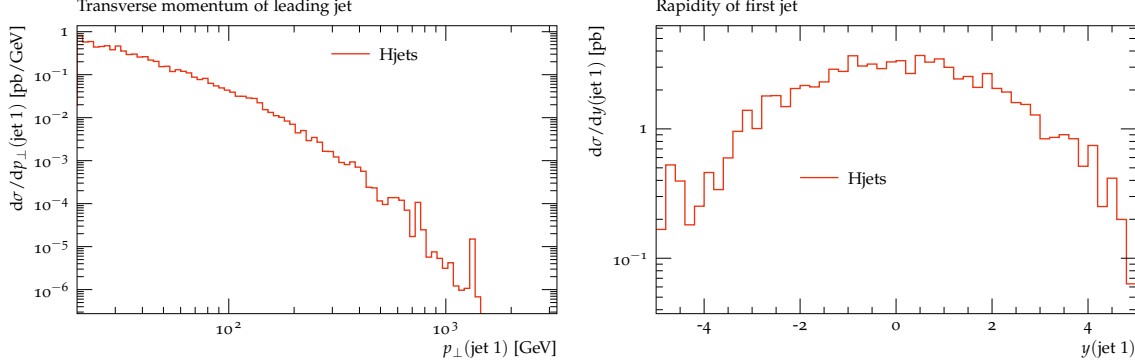

Figure 4: Hardest jet transverse momentum and rapidity for the production of a Higgs boson together with between one and three jets.

### 3.2.1. Quark mass corrections

For accurate predictions in the high-energy region, we have to take into account the finite top quark mass. In the following, we assume a mass of $174\,\text{GeV}$. On the SHERPA side, we can add AMEGIC [26] and OPENLOOPS [27, 28] to ME_SIGNAL_GENERATOR and insert the following lines into the (run) block:

```
# finite top mass effects
KFACTOR GGH
OL_IGNORE_MODEL 1
OL_PARAMETERS preset 2 allowed_libs pph2,pphj2,pphjj2 psp_tolerance 1.0e-7
```

HEJ needs to be compiled with support for QCDLOOP [29] to incorporate quark mass corrections in Higgs boson production. We can include them by adding

```
Higgs coupling:
   use impact factors: false
   mt: 174
```

to the configuration file.

For higher jet multiplicities, we face the problem that it is no longer feasible to compute the leading-order input with exact dependence on the top quark mass $m_t$. However, we can still retain this dependence and also include the dependence on the bottom-quark mass $m_b$ in the high-energy resummation.

As in equation (1), the weight $w$ of a leading-order matched resummation event has the following dependence on the leading-order matrix element $\mathcal{M}_{\mathrm{LO}}$ and the all-order HEJ matrix element $\mathcal{M}_{\mathtt{HEJ}}(m_b, m_t)$:

$$w \propto \frac{|\mathcal{M}_{\mathrm{LO}}(m_b, m_t)|^2 |\mathcal{M}_{\mathtt{HEJ}}(m_b, m_t)|^2}{|\mathcal{M}_{\mathtt{HEJ,LO}}(m_b, m_t)|^2}. \tag{20}$$

For consistency, the values for the quark masses have to match those used in $\mathcal{M}_{\mathrm{LO}}$. Therefore, if the leading-order input is only known for $m_b \to 0, m_t \to \infty$ the correct reweighting factor is

$$w \propto \frac{|\mathcal{M}_{\mathrm{LO}}(0, \infty)|^2 |\mathcal{M}_{\mathtt{HEJ}}(m_b, m_t)|^2}{|\mathcal{M}_{\mathtt{HEJ,LO}}(0, \infty)|^2} = \frac{|\mathcal{M}_{\mathrm{LO}}(0, \infty)|^2 |\mathcal{M}_{\mathtt{HEJ}}(m_b, m_t)|^2}{|\mathcal{M}_{\mathtt{HEJ,LO}}(m_b, m_t)|^2} \times \frac{|\mathcal{M}_{\mathtt{HEJ,LO}}(m_b, m_t)|^2}{|\mathcal{M}_{\mathtt{HEJ,LO}}(0, \infty)|^2}. \tag{21}$$

Currently, there is no built-in HEJ option for choosing different quark mass values in $\mathcal{M}_{\mathtt{HEJ}}$ and $\mathcal{M}_{\mathtt{HEJ,LO}}$. However, HEJ supports flexible custom analyses, which allow us to manually reweight by the correction factor $|\mathcal{M}_{\mathtt{HEJ,LO}}(m_b, m_t)|^2 / |\mathcal{M}_{\mathtt{HEJ,LO}}(0, \infty)|^2$ in equation (21). We can also use this opportunity to include bottom quark mass corrections in the case where only the exact leading-order dependence on the top quark mass is available.

Custom analyses are described in the HEJ user documentation on https://hej.hepforge.org, where also a template is provided. The reweighting can be implemented as shown here:

higgs_matching_analysis.cc

```cpp
#include <string>
#include <memory>

#include "HEJ/Analysis.hh"
#include "HEJ/Config.hh"
#include "HEJ/event_types.hh"
#include "HEJ/Event.hh"
#include "HEJ/HiggsCouplingSettings.hh"
#include "HEJ/MatrixElement.hh"
#include "HEJ/RivetAnalysis.hh"
#include "HEJ/YAMLreader.hh"

#include "yaml-cpp/yaml.h"

namespace LHEF {
  class HEPRUP;
}
```

```cpp
namespace {
  // Dummy function for alpha_s(mu)
  // Since the dependence on alpha_s cancels out in the matrix element ratio
  // we can return any (finite, non-zero) value we like
  double alpha_s_dummy(double /* mu */) {
    return 1.;
  }

  // Helper function to initialise modulus square of HEJ matrix element
  HEJ::MatrixElement create_HEJ_ME(
    YAML::Node const & yaml_config,
    std::string const &  Higgs_coupling_name
  ) {
      HEJ::MatrixElementConfig config;
      config.log_correction = false;
      config.ew_parameters = HEJ::get_ew_parameters(yaml_config);
      config.Higgs_coupling = HEJ::get_Higgs_coupling(
        yaml_config,
        Higgs_coupling_name
      );
      return HEJ::MatrixElement{alpha_s_dummy, config};
  }

  // Check if we perform resummation for an event
  bool is_resummable(HEJ::Event const & event) {
    switch(event.type()) {
    case HEJ::event_type::FKL:
    case HEJ::event_type::unordered_backward:
    case HEJ::event_type::unordered_forward:
      return true;
    default:
      return false;
    }
  }

  // Our custom analysis
  class HiggsMatchingAnalysis: public HEJ::Analysis {
    public:
      HiggsMatchingAnalysis(
        YAML::Node const & config,
        LHEF::HEPRUP const & heprup
      ):
        me_exact_{create_HEJ_ME(config, "Exact Higgs coupling")},
        me_approx_{create_HEJ_ME(config, "LO Higgs coupling")},
        rivet_analysis_{config["Rivet analysis"], heprup}
    {}

      void fill(
        HEJ::Event const & ev,
        HEJ::Event const & LO_event
      ) override {
        auto event = ev;
        if(is_resummable(event)) {
          // Compute ratio of exact and approximate matrix element squares
          // truncated to leading order.
          // In HEJ, we split the squares of tree-level matrix elements into a
          // scale-dependent "parametric" factor containing the couplings and a
          // scale-independent "kinetic" factor. Since the dependence on the
          // couplings cancels exactly in the ratio, we only need the latter
```

```cpp
        // part 'tree_kin' here.
        // To account for the possibility of interference, 'tree_kin' returns
        // a 'std::vector' instead of a single value. Here, the 'std::vector'
        // has only a single element.
        const double me_exact = me_exact_.tree_kin(LO_event).front();
        const double me_approx = me_approx_.tree_kin(LO_event).front();
        const double reweight = me_exact / me_approx;
        event.central().weight *=  reweight;
        // If we perform scale variation we also have to rescale all other
        // weights
        for(auto & var: event.variations()) {
          var.weight *= reweight;
        }
      }
      // Pass the potentially reweighted event to Rivet
      rivet_analysis_.fill(event, LO_event);
    }

    bool pass_cuts(
      HEJ::Event const & /* event */,
      HEJ::Event const & /* LO_event */
    ) override {
      return true;
    }

    void set_xs_scale(double scale) override {
      rivet_analysis_.set_xs_scale(scale);
    }

    void finalise() override {
      rivet_analysis_.finalise();
    }

  private:
    HEJ::MatrixElement me_exact_;
    HEJ::MatrixElement me_approx_;
    HEJ::RivetAnalysis rivet_analysis_;
  };
}

extern "C"
__attribute__((visibility("default")))
std::unique_ptr<HEJ::Analysis> make_analysis(
    YAML::Node const & config, LHEF::HEPRUP const & heprup
){
  return std::make_unique<HiggsMatchingAnalysis>(config, heprup);
}
```

We can then compile the analysis into a shared object library with a compiler supporting
C++17, for instance a recent version of g++:

```
g++ -Wall -Wextra $(HEJ-config --cxxflags) -fPIC -shared -O2 \
  -fvisibility=hidden \
  -Wl,-soname,libhiggs_matching_analysis.so \
  -o libhiggs_matching_analysis.so higgs_matching_analysis.cc
```

To use our custom analysis, we adjust the HEJ configuration file. We use YAML anchors
(starting with &) and references (starting with *) to ensure that the settings passed to

the analysis are consistent. The following code listing is for the case of a leading-order prediction in the infinite top-quark mass limit.

config_Hjets_mbmt.yml

```
trials: 10

resummation jets:
  min pt: 20
  algorithm: antikt
  R: 0.4

fixed order jets:
  min pt: 15

event treatment:
  FKL: reweight
  unordered: reweight
  extremal qqbar: keep
  central qqbar: keep
  non-resummable: keep

scales: H_T/2

vev: 246.2196508

particle properties: &particle_properties
  Higgs:
    mass: 125
    width: 0.004165
  W:
    mass: 80.385
    width: 2.085
  Z:
    mass: 91.1876
    width: 2.4952

random generator:
  name: mixmax
  seed: 1

Higgs coupling: &exact_h_coupling
    use impact factors: false
    mt: 174
    include bottom: true
    mb: 2.8

analyses:
  - plugin: ./libhiggs_matching_analysis.so
    Exact Higgs coupling: *exact_h_coupling
    LO Higgs coupling:
      use impact factors: true
    Rivet analysis:
      rivet: MC_JETS
      output: H1j
    vev: 246.2196508
    particle properties: *particle_properties

log correction: false
```

If the leading order prediction includes the exact dependence on the top-quark mass, but not the dependence on the bottom-quark mass, one should replace the `LO Higgs coupling` entry by

```
    LO Higgs coupling:
      use impact factors: false
      mt: 174
```

Table 6 lists reference cross sections and figure 5 shows corresponding distributions combining samples with up to three jets.

| Run | $H + 1$ jet | $H + 2$ jets | $H + 3$ jets |
|---|---|---|---|
| SHERPA | $25.911 \pm 0.442621$ | see tab. 5 | see tab. 5 |
| HEJ | $11.50 \pm 0.2616$ | $4.409 \pm 0.1947$ | $1.648 \pm 0.1544$ |

Table 6: Reference cross sections in picobarn for Higgs boson plus jets production with finite quark mass effects. All `HEJ` resummation results account for massive bottom and top quarks. The SHERPA prediction for $H + 1$ jet uses the physical top-quark mass, but assumes a massless bottom quark. The fixed-order predictions for higher multiplicities do not include finite quark mass effects.

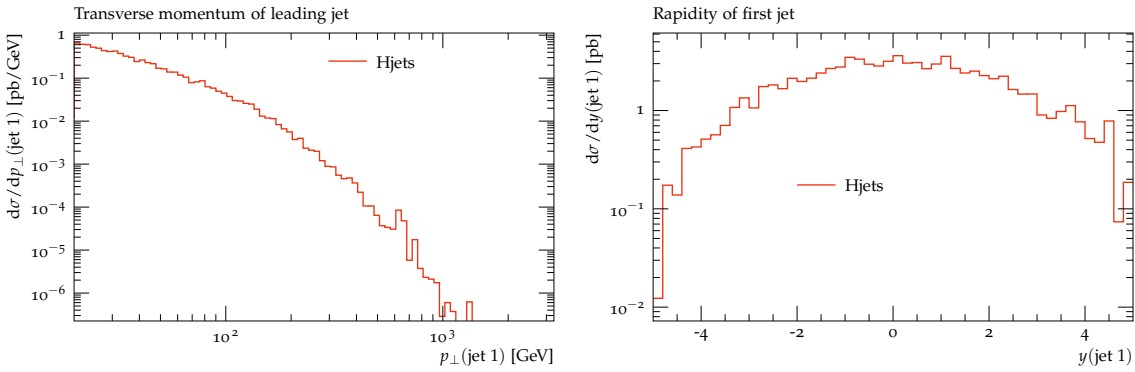

Figure 5: Hardest jet transverse momentum and rapidity for the production of a Higgs boson together with between one and three jets including finite quark mass effects.

### 3.2.2. Matching to next-to-leading order

Using `HEJ` 2.2, we can extend the fixed-order accuracy of distributions from LO to NLO, cf. section 2.5. In the following, we consider NLO matching for the production of a Higgs boson together with at least one jet. While we derive the matching in the limit of an infinitely heavy top quark, the resulting rescaling factors can also be used to improve HEJ predictions incorporating quark-mass corrections. We apply the matching bin-by-bin in the resulting histograms. We can generate an NLO prediction using SHERPA and OPENLOOPS by adjusting the run card as follows.

Run.dat

```
(run){
  EVENTS 10000
```

```
  MODEL HEFT
  USE_PDF_ALPHAS 1
  MASS[25] 125
  WIDTH[25] 0.004165

  MASSIVE[4] 0
  YUKAWA[4] 0.
  MASS[4] 0.

  MASSIVE[5] 0
  YUKAWA[5] 0.
  MASS[5] 0.

  BEAM_ENERGY:=6500.
  BEAM_1 2212 BEAM_ENERGY
  BEAM_2 2212 BEAM_ENERGY

  PDF_LIBRARY LHAPDFSherpa
  PDF_SET CT14nlo

  SCALES VAR{H_T2/4}

  EVENT_GENERATION_MODE Weighted
  ME_SIGNAL_GENERATOR Comix OpenLoops

  FRAGMENTATION Off
  YFS_MODE 0
  MI_HANDLER None
  CSS_MAXEM 0
  BEAM_REMNANTS 0

  ANALYSIS Rivet
  ANALYSIS_OUTPUT Hj_NLO
}(run)

(processes){
  Process 93 93 -> 25 93
  Order (*,0,1)
  NLO_QCD_Mode Fixed_Order
  NLO_QCD_Part BVIRS
  Integration_Error 0.02
  Loop_Generator OpenLoops
  End process
}(processes)

(selector){
  FastjetFinder antikt 1 20. 0 0.4
}(selector)

(analysis){
  BEGIN_RIVET {
  USE_HEPMC_SHORT 1
  IGNOREBEAMS 1
  -a MC_JETS
 } END_RIVET
}(analysis)
```

The resulting cross section is $(18.4425 \pm 3.83924)$ pb.

For the HEJ prediction, we add the option

```yaml
NLO truncation:
  enabled: true
  nlo order: 1     # number of jets
```

to the configuration file `config_Hjets.yml` from section [3.2], change the name of the RIVET output file to `Hj_HEJ_NLO_1j.yoda`, and run `HEJ` on the previously generated leading-order input file assuming an infinitely heavy top quark. We obtain a cross section of $(7.846 \pm 1.091)$ pb. The result is *exclusive* in the number of jets; in contrast to the SHERPA prediction the contribution from events with two jets is not included, yet. To generate the missing piece, we remove the NLO process configuration from the above SHERPA run card, i.e. we change the `ANALYSIS_OUTPUT` to `Hjj_LO`, remove OPENLOOPS from `ME_SIGNAL_GENERATOR` and delete the following lines.

```
NLO_QCD_Mode Fixed_Order
NLO_QCD_Part BVIRS
Integration_Error 0.02
Loop_Generator OpenLoops
```

We then increase the number of jets to two and run SHERPA to generate a file `Hjj_LO.yoda`, with a total cross section of $(8.80745 \pm 0.321808)$ pb. We add this contribution to the truncated `HEJ` result:

```
yodastack -o Hj_HEJ_NLO.yoda Hj_HEJ_NLO_1j.yoda Hjj_LO.yoda
```

To obtain the final NLO matched prediction, we should multiply each histogram in the original `HEJ` output by the ratio of the corresponding histograms in `Hj_NLO.yoda` and `Hj_HEJ_NLO.yoda`. The following script gives an example of how this reweighting can be implemented. For the sake of brevity we have omitted the error handling code, which is of course essential in actual applications.

reweight_NLO.py

```python
#!/usr/bin/env python

import yoda

nlo = yoda.read("Hj_NLO.yoda")
hej_nlo = yoda.read("Hj_HEJ_NLO.yoda")
full = yoda.read("Hjets.yoda")
full_rescaled = {}

for path, full in full.items():
    try:
        reweight_fact_scatter = hej_nlo[path] / nlo[path]
        reweight_fact = yoda.core.Histo1D(path, full.title())
        reweight_fact.addBins(full.xEdges())
        for p in reweight_fact_scatter.points():
            reweight_fact.fill(p.x(), p.y())
        full_rescaled[path] = full / reweight_fact
        full_rescaled[path].setPath(path)
    except:
        pass

yoda.write(full_rescaled, "Hjets_rescaled.yoda")
```

Generating `Hjets.yoda` as described in section 3.2.1, the resulting NLO-matched rapidity distribution of the hardest jet is shown in figure 6. Note that the MC_JETS RIVET analysis employs adaptive binning for a number of distributions, causing the division of the respective histograms to fail. This problem can of course be circumvented by using a custom analysis.

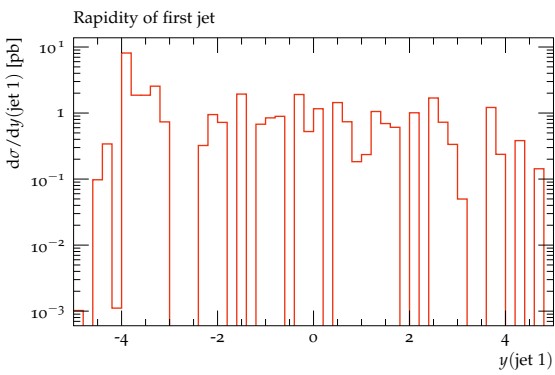

Figure 6: Differential NLO matching for the rapidity distribution of the hardest jet produced together with a Higgs boson.

## 3.3. Charged lepton pair production with many jets

The production of two charged leptons with jets was first implemented in HEJ 2.1 [30] for at most moderate jet multiplicities, where exact fixed-order matching is feasible. To overcome this difficultly, in HEJ 2.2 approximate high-multiplicity events can be generated with the HEJ Fixed Order Generator.

In the following example, we consider the process $pp \to \mu^+\mu^- + \geq 2$ jets with the same parameters as in previous examples, see table 2. We assume that predictions including up to 4 jets have already been produced as described in sections 3.1 and 3.2. To generate input events with 5 jets, we adapt the configuration file `configFO.yml` distributed together with HEJ:

configFO_Zjets.yml

```
## Number of generated events
events: 10000

jets:
  min pt: 15        # minimal jet pt, should be slightly below analysis cut
  peak pt: 20       # peak pt of jets, should be at analysis cut
  algorithm: antikt # jet clustering algorithm
  R: 0.4            # jet R parameter
  max rapidity: 5   # maximum jet rapidity

## Particle beam
beam:
  energy: 6500
  particles: [p, p]

## PDF ID according to LHAPDF
pdf: 13100
```

```
## Scattering process
process: p p => mu mu_bar 5j

## Fraction of events with two extremal emissions in one direction
## that contain an subleading emission e.g. unordered emission
subleading fraction: 0.05

## Allow different subleading configurations.
## By default all subleading channels are disabled.
subleading channels:
  - unordered

## Unweighting parameters
## remove to obtain weighted events
unweight:
  sample size: 10000  # should be similar to "events:", but not more than ~10000
  max deviation: 0

## Central scale choice
scales: H_T / 2

## Event output files
event output:
  - events_Z5j.lhe

## Selection of random number generator and seed
random generator:
  name: mixmax
  seed: 1

## Vacuum expectation value
vev: 246.2196508

## Properties of the weak gauge bosons
particle properties:
  Higgs:
    mass: 125
    width: 0.004165
  W:
    mass: 80.385
    width: 2.085
  Z:
    mass: 91.1876
    width: 2.4952
```

By setting the `peak pt` option we ensure that most events are generated above the analysis cut of 20 GeV. This means that there is no need for two separate runs with different transverse momentum cuts as described in section 3.1.3 for conventional fixed-order generators. We can now generate events with

```
docker run -v $PWD:$PWD -w $PWD hejdock/hej HEJFOG configFO_Zjets.yml
```

when using the HEJ Docker container or

```
HEJFOG configFO_Zjets.yml
```

when using a local HEJ installation. We obtain a cross section of $(11.53 \pm 1.607)$ pb. Afterwards, we can run HEJ on the resulting `events_Z5j.lhe` event file to obtain resummed

predictions for $pp \to \mu^+\mu^- + 5$ jets. These could then be combined with the results for lower multiplicities as described in section 3.1.2.

## 4. Conclusions

The `HEJ` Monte Carlo event generator provides accurate high-energy descriptions for a steadily growing range of scattering processes. The new version 2.2 adds predictions for the QCD corrections to the production of two W bosons with the same charge together with two or more jets, which are pivotal for experimental measurements of weak boson fusion. A further major improvement concerns the gluon-fusion production of a Higgs boson with jets, where for the first time final states with only a single jet are included in the description.

While LO-matched predictions for charged lepton pair production with jets were already available in `HEJ` 2.1, the new release allows to supplement these with LL-accurate high-energy corrections for high multiplicities where exact LO generation may no longer be feasible. Furthermore, `HEJ` 2.2 allows NLO matching at the level of differential distributions and facilitates a computationally more efficient event input generation by separating off low-$p_\perp$ events with a small contribution to the final predictions.

We have given detailed examples for the new features. The program code as well as comprehensive documentation including options added in previous versions are available on https://hej.hepforge.org.

## Acknowledgements

HEJ uses `FastJet` [31] and `LHAPDF` [32].

We are grateful to the other members of the `HEJ` collaboration for helpful comments and discussions. We are pleased to acknowledge funding from the UK Science and Technology Facilities Council, the Royal Society, and the ERC Starting Grant 715049 "QCD-forfuture". For the purpose of open access, the authors have applied a Creative Commons Attribution (CC BY) licence to any Author Accepted Manuscript version arising from this submission.

# Appendix A. Download and installation

Detailed documentation for `HEJ 2.2` can be found on [https://hej.hepforge.org/](https://hej.hepforge.org/). In the following, we briefly explain how to download and install the program.

## *Appendix A.1. Download*

A tar archive of the `HEJ 2` source code can be downloaded and decompressed with the command:

```
curl https://hej.hepforge.org/downloads?f=HEJ_2.2.tar.gz | tar -xz
```

To obtain the latest stable `HEJ` version, `HEJ_2.2.tar.gz` should be replaced by `HEJ.tar.gz`.

Alternatively, the `HEJ` source code can be obtained by installing the git version control system [33] and running:

```
git clone https://phab.hepforge.org/source/hej.git
```

We also provide a Docker image containing a `HEJ 2` installation (including the HEJ Fixed Order Generator). This image can be pulled with:

```
docker pull hejdock/hej
```

When using the Docker image the remaining installation steps can be skipped.

In addition to working with Docker, these images will also work with Apptainer/SingularityCE [34, 35]. This comes with a caveat that you may need to source the software inside the image before running `HEJ`:

```
source /cvmfs/pheno.egi.eu/HEJV2/HEJ_env.sh
```

Users can also make use of the MPI libraries included in the images for efficient fixed order event generation for `HEJ` input using Sherpa which is configured with MPI support. Note that `HEJ` itself does not provide support for MPI.

## *Appendix A.2. Prerequisites*

Before installing `HEJ 2`, you need the following programs and libraries:

- CMake [36] version 3.1

- A compiler supporting the C++17 standard, for example gcc 7 or later [37]

- FastJet [31]

- CLHEP [38] version 2.3

- LHAPDF [32] version 6

- The IOStreams and uBLAS boost [39] libraries

- yaml-cpp [40]

- `autoconf` and `automake` for FORM [18]

In addition, some optional features have additional dependencies:

- Version 2 of QCDLoop [29] is required to include finite top mass corrections in Higgs boson + jets production.

- HepMC [41] versions 2 and 3 enable event output in the respective format.

- Rivet [22] version 3.1.4 or later together with HepMC 2 or 3 allow using Rivet analyses.

- HighFive[42] has to be installed in order to read and write event files in the HDF5 [43]-based format suggested in [44].

We strongly recommend to install these programs and libraries to standard locations:

- The executable files should be inside one of the directories listed in the PATH environment variable. This concerns cmake, the C++ compiler, and the executables contained in autoconf and automake.

- The library header files ending with .h, .hh, or .hpp should be in a directory where they are found by the C++ compiler. For gcc or clang, custom locations can be specified using the CPLUS_INCLUDE_PATH environment variable.

- The compiled library files ending with .a, .so, or .dylib should be in a directory where they are found by the linker. Custom locations can be set via the LIBRARY_PATH environment variable. For shared object libraries (.so or .dylib) custom locations should also be part of LD_LIBRARY_PATH on linux and DYLD_FALLBACK_LIBRARY_PATH or DYLD_LIBRARY_PATH on macOS.

*Appendix A.3. Compilation*

To compile and install HEJ 2 run:

```
cmake source/directory -DCMAKE_INSTALL_PREFIX=target/directory
make install
```

source/directory is the directory containing the file CMakeLists.txt. If you omit -DCMAKE_INSTALL_PREFIX=target/directory, HEJ 2 will be installed to some default location.

In case some of the aforementioned prerequisites are not found by cmake you can give a hint by adding an additional argument -Dlibname_ROOT_DIR=/directory/with/library, where libname should be replaced by the name of the library in question. For example, if FastJet is installed in the subdirectory .local of your home directory with the libfastjet.* library files in .local/lib and the header files ending with .hh in .local/include/fastjet you can pass -Dfastjet_ROOT_DIR=$HOME/.local to cmake.

If cmake fails to find (the correct) boost path, try setting -DBOOST_ROOT=/path/to/boost, this will force cmake to search for boost only in /path/to/boost.

To not include specific packages one can add -DEXCLUDE_packagename=TRUE to cmake, e.g. by setting -DEXCLUDE_rivet=TRUE HEJ 2 will not be interfaced to Rivet even if it is available on the system.

*Appendix A.4. Testing*

To test your installation, download the NNPDF 2.3 PDF set with:

```
lhapdf install NNPDF23_nlo_as_0119
```

and run:

```
make test
```

The test data of `HEJ` are stored in a Git Large File Storage [45] format. `git clone` therefore requires `git-lfs` to download the data correctly.

*Appendix A.5. Running*

To run `HEJ`, use the command

```
HEJ config.yml events.lhe
```

where `config.yml` is a configuration file and `events.lhe` a Les Houches event file [23]. Explicit examples are given in section 3.

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
