# Peer review of "HEJ 2.2: W boson pairs and Higgs boson plus jet production at high energies"

_SciPost Physics Codebases_

## Round 1 · Referee Report · Anonymous (Referee 1) · 2023-7-7

Strengths

1) The draft is well written.

2) The program allows for the simulation of the phenomenologically important W-W-jj and Hj production processes.

3) The authors provide useful examples to illustrate the code usage.

Weaknesses

1) The physics of the processes implemented in the code has already been published separately.

2) The draft does not explicitly contain general instructions for downloading, installing and running the software as requested by SciPost acceptance criteria.

3) No benchmark numbers are quoted.

Report

In their article "HEJ 2.2: W boson pairs and Higgs boson plus jet production at high energies" the authors present an updated version of the HEJ Monte Carlo event generator for the simulation of hadronic scattering processes at high energies. Compared to previous versions of the program, the following features have been added: The same-sign W boson production process in association with two or more jets; the Higgs production process with at least one jet; a new prediction for charged lepton-pair production with high jet multiplicities. Furthermore, some technical improvements have been made in the interface to fixed-order generators.

The article is well written and clearly describes the features of the presented program. For each of the new processes, the key aspects of the implementation are described, and an example for the usage of the program is given. The sample run cards and configuration files are useful for readers planning to use the code themselves.

The added value compared to existing software, in particular older versions of the HEJ program, is clearly described. In particular, the phenomenologically important W-W-jj and Hj production processes are included in the new program version.

SciPost requires that benchmarking tests be provided. Apart from Fig.2 no explicit results are shown in the draft. It might be helpful for future users of the code to have a few explicit benchmark numbers at hand to be able to test the correct usage of the program.

SciPost also requires a complete documentation including detailed instructions on downloading, installing and running the software (see SciPost acceptance criteria). While detailed references to particular aspects of the example processes discussed in the draft are given, in the text I did not find a general documentation for installing and running the program. Such instructions can be found on the hepforge page of the program. Nontheless, to comply with SciPost's requirements I'd suggest that the authors include more details on the general usage of the program in the draft (for instance by adding an appendix).

Requested changes

1) I find the notation introduced in Sec.2.1 for the configuration of the pure multijet production process and later adapted to other processes slightly difficult to get used to. The authors use fa,fb to denote the flavors of both incoming and outgoing partons, but then introduce the notation p1,...,pn for the momenta of the final-state partons. The reader not used to this notation may wonder which momenta correspond to the outgoing partons fa,fb. Are they pa,pb? Or either of the p1,...,pn?

2) For the same-sign-W production process in Sec. 3.1.1 the bottom quark mass is explicitly set to zero. In the printed part of the Run.dat file no statements are made on the number of active flavors and the treament of the top quark. However, unless restrictions on such contributions are made, sub-processes with incoming b-quarks result in outgoing top quarks. How is the mass of the top quark treated in such contributions?

3) The quantity H_T that is frequently used as a scale is never defined in the text.

4) In Fig. 2 (left) labels on the y axis are missing.

  • validity: good
  • significance: good
  • originality: ok
  • clarity: good
  • formatting: good
  • grammar: excellent

Author:  Andreas Maier  on 2023-08-08  [id 3884]

(in reply to Report 1 on 2023-07-07)

We thank the referee for their report and helpful suggestions that we would be happy to address in a revised version of our manuscript. Responding firstly to the numbered requests:

1) Indeed we use the notation p1, ..., pn for the momenta of final-state partons. For leading-logarithmic configurations the parton with momentum p1 has flavour fa and the parton with momentum pn has flavour fb. For next-to-leading-logarithmic configurations this is in general no longer the case. We would be happy to add a sentence clarifying this point.

2) Our description limits the number of active flavours to 4. We agree we should explain this explicitly in the text and would be happy to add that to section 2.2. We have verified that Sherpa 2.2 with the provided run card does not produce events with incoming bottom (and outgoing top) quarks.

3) Indeed, we omitted to define H_T and would be happy to add this where it is first used in section 3.1.

4) We wanted to show the output as the user would obtain it with the lines of code provided (and in particular linked to a publicly available rivet analysis as opposed to a custom analysis of our own). Unfortunately the plot in the left of Fig. 2 came out without numerical labels for the ticks. We are happy to add some for clarity and comment in the text that we have added that by hand.

Responding to the other suggestions:

  • The plots in Fig. 2 give examples of output for users to check against, but we would be happy to add further benchmark points in a modified version. Specifically we would give cross sections (and errors) obtained for each process discussed.

  • The referee also suggested it would be better if the full download instructions from hepforge were included in an appendix. We would be happy to do that.

---

## Round 1 · Referee Report · Anonymous (Referee 2) · 2023-7-14

Report

This manuscript discusses a set of new features of the HEJ computer program. In particular, the novelties are the possibility to compute differential cross sections in with high-energy resummation for same-sign W-boson pair production, and the production of a Higgs boson with a single jet. Minor improvements to the prediction of a lepton pair, the matching to NLO computations and the treatment of small transverse momentum jets are also described. The manuscript describes the new features well, and includes plenty of practical information on how to use the new features of the code, and is properly referenced. However, the two main new features have already their own dedicated publications (ref. [21] and [14], respectively), and rather similar examples have already been given in ref. [29]. Therefore, this manuscript does not contain a significant amount of original research, and I cannot recommend this paper for publication.

Also, figure 2 is not correct. There are no labels on the y-axis of the left plot, and the total cross section for same sign W-boson pair production (including decays to a single lepton flavour, and a 20 GeV jet pT cut) should be on the order of several fb's. The integral of the distribution in the right plot, is much smaller than that and therefore cannot be correct.
  • validity: -
  • significance: -
  • originality: -
  • clarity: -
  • formatting: -
  • grammar: -

Author:  Andreas Maier  on 2023-08-08  [id 3885]

(in reply to Report 2 on 2023-07-14)

We thank the referee for their report and comments. However we don't agree with their statement that this manuscript is not suitable for publication in SciPost Physics Codebase on the grounds that it does not contain a significant amount of original research. Indeed there have been publications for the same-sign W-boson pair production and Higgs boson production with a single jet previously, and their more detailed physics analysis is not repeated here. This manuscript is a SciPost Physics Codebase submission, not a general SciPost Physics submission. It represents the first public release of code capable of describing the two processes above at leading logarithmic accuracy in s-hat/p_perp^2, where these logarithms are important in the regions probed by current LHC analyses. As noted by the first referee, these are phenomenologically important. This fits in the scope of SciPost Physics Codebase, which includes "Libraries providing new or significantly improved components or interfaces which enhance the capability, performance, or productivity of scientific software".

The referee also questions the accuracy of the results in Fig. 2. The original output from the HEJ generation was indeed of the order of a few fb before passing through the chosen public rivet analysis. After this, the cross section drops significantly due to their additional acceptance cuts on leptons, missing pT and additional isolation cuts. We have observed that the same significant drop also occurs for a LO sample. After a suggestion in the first report, we have suggested that we will include benchmark numbers for the total cross section after generation and before analysis for both processes. We think this will also clarify this point.

---

## Round 1 · Referee Report · Anonymous (Referee 3) · 2023-8-24

Report

This paper describes an update of the HEJ code that adds functionality for additional processes such as Higgs plus one jet and same-sign W-bosons plus jets. HEJ implements resummation of large s/pT^2 logs in processes with high energy jets and rapidity gaps, an important component of the LHC program. The intro of the paper gives a succinct but clear description of the issue and why it is interesting. The paper summarizes all the relevant formula and does an excellent job of describing how to set input files in example runs for the code. I recommend it for publication after the authors consider the following requests.

Requested changes

(1) Fig. 2 left plot: please label the y-axis.
(2) In general more benchmark numbers and plots are needed. Users need to know if they are running the code correctly. Particularly since H+1 or more jets is new I would recommend sample numbers and plots for this process. For both this and same-sign W production some tables of cross sections are needed.
(3) For section 3.1.3, since the authors emphasize the usefulness of the separate low-statistics low-pT run both here and in section 2.4, and this is primarily a code description paper, I would recommend an example of showing how this and the inefficient approach agree for 1-2 sample distributions. This can be incorporated as part of the additional benchmarking requested.

---

## Editorial Decision

resubmitted